# Updates on Immunotherapy and Immune Landscape in Renal Clear Cell Carcinoma

**DOI:** 10.3390/cancers13225856

**Published:** 2021-11-22

**Authors:** Myung-Chul Kim, Zeng Jin, Ryan Kolb, Nicholas Borcherding, Jonathan Alexander Chatzkel, Sara Moscovita Falzarano, Weizhou Zhang

**Affiliations:** 1Department of Pathology, Immunology and Laboratory Medicine, University of Florida, Gainesville, FL 32610, USA; my.kim@ufl.edu (M.-C.K.); Zengjin@ufl.edu (Z.J.); Ryankolb@ufl.edu (R.K.); sfalzarano@ufl.edu (S.M.F.); 2UF Health Cancer Center, University of Florida, Gainesville, FL 32610, USA; 3Department of Pathology and Immunology, Washington University, St. Louis, MO 63110, USA; borcherding.n@wustl.edu; 4Department of Medicine Hematology and Oncology Division, University of Florida, Gainesville, FL 32610, USA; Jonathan.Chatzkel@medicine.ufl.edu

**Keywords:** single-cell RNA sequencing, immune landscape, cancer immunotherapy, clear cell renal cell carcinoma

## Abstract

**Simple Summary:**

Clear cell renal cell carcinomas (ccRCC) have several distinct immunological features, including a high degree of immune infiltration and relatively low mutational burdens, the resistance to cytotoxic chemotherapy, and relative sensitivity to anti-angiogenic therapy and immunotherapies. Immune checkpoint inhibitor (ICI) therapy has become standard care in the treatment of ccRCC, but a better understanding of the molecular and cellular characteristics of ccRCC is needed to truly optimize the use of ICI therapy. With a focus on cancer immunology, we summarize the clinical trials of ICIs in ccRCC, the molecular and cellular correlates of these clinical trials, and the single-cell RNA sequencing studies to provide a comprehensive overview of the immune landscape within the ccRCC tumor microenvironment, in particular in the context of ICI therapy. We will discuss potential molecular and cellular biomarkers that can be used to predict therapeutic responses in ccRCC patients.

**Abstract:**

Several clinicopathological features of clear cell renal cell carcinomas (ccRCC) contribute to make an “atypical” cancer, including resistance to chemotherapy, sensitivity to anti-angiogenesis therapy and ICIs despite a low mutational burden, and CD8^+^ T cell infiltration being the predictor for poor prognosis–normally CD8^+^ T cell infiltration is a good prognostic factor in cancer patients. These “atypical” features have brought researchers to investigate the molecular and immunological mechanisms that lead to the increased T cell infiltrates despite relatively low molecular burdens, as well as to decipher the immune landscape that leads to better response to ICIs. In the present study, we summarize the past and ongoing pivotal clinical trials of immunotherapies for ccRCC, emphasizing the potential molecular and cellular mechanisms that lead to the success or failure of ICI therapy. Single-cell analysis of ccRCC has provided a more thorough and detailed understanding of the tumor immune microenvironment and has facilitated the discovery of molecular biomarkers from the tumor-infiltrating immune cells. We herein will focus on the discussion of some major immune cells, including T cells and tumor-associated macrophages (TAM) in ccRCC. We will further provide some perspectives of using molecular and cellular biomarkers derived from these immune cell types to potentially improve the response rate to ICIs in ccRCC patients.

## 1. Introduction

Renal cell carcinomas (RCC) arise from the renal epithelium and account for more than 90% of cancers occurring in the kidney [1]. There are about 76,000 new cases annually in the U.S. and 403,000 worldwide, accounting for about 3% of all cancers [2,3]. About 70% of patients with RCC have localized tumors at the time of diagnosis, and 12% of the cancer patients have metastatic tumors [4]. Approximately 50% of patients with localized RCC ultimately develop metastatic disease, and the 5-year survival rate of patients with metastatic RCC is approximately 14% [1,5,6]. In general, about 25% to 50% of patients with primary RCC experience recurrence following nephrectomy after five years [7]. RCC is histologically classified into subtypes, of which clear cell RCC (ccRCC) is the most common–accounting for more than 80% of RCCs, followed by papillary RCC and chromophobe RCC [1,4]. ccRCC is characterized by the abundance of glycogen and lipids in the cytosol [1,4]. Most patients with ccRCC show chromosomal 3p loss and genomic mutations in the *Von Hippel-Lindau Tumor Suppressor* (*VHL*) allele [8], followed by secondary loss of multiple tumor suppressor genes, including *PBRM1*, *SETD2*, *BAP1,* and/or *KDM5C* [9]. The VHL inactivation stabilizes hypoxia-inducible factors (HIFs) in ccRCC, including HIF1α and HIF2α [10]. The activation of HIFs leads to transcriptional activation of numerous HIF target genes, including vascular endothelial growth factor (*V**EGF*), which is one of the major known mechanisms responsible for high angiogenesis and inflammatory response in the ccRCC tumor microenvironment [10,11].

Tyrosine kinase inhibitors (TKIs) are representative first-line anti-angiogenic targeted therapies to inhibit VEGF and its receptor (VEGFR) signaling in patients with metastatic ccRCC. These TKIs are effective, with a limited number of patients showing complete remission of ccRCC [12]. Generally, however, these targeted therapies are only palliative, and the utility of this therapy is frequently limited by drug resistance [13].

The Food and Drug Administration (FDA) approved the use of nivolumab (anti-PD-1) for patients with RCC in 2015. Since then, numerous clinical trials have demonstrated the safety and efficacy of a variety of immune checkpoint inhibitors (ICI) for RCC patients [14,15]. Spontaneous immune activation is thought to contribute to the regression of 1 to 7% of ccRCC patients [16,17,18,19]. Early clinical trials enhancing T cell proliferation through high-dose interleukin 2 (IL-2) achieved up to 20% of therapeutic response [20]. ICI monotherapy showed 25 to 42% response rates in ccRCC patients [15,21]. In studies evaluating ICI in combination with anti-VEGF or TKIs as a first-line therapy, it significantly improved the clinical outcome in patients with ccRCC, showing an objective response rate (ORR) of 50 to 59%, including 4 to 12% complete response (CR) rates, depending on experimental settings [21,22,23,24,25]. Meanwhile, phase III clinical trials investigating ICI in combination with TKIs reported 48% to 82% of treatment-related adverse events with grade 3 or higher [22,23,25,26,27]. Safety evaluation reveals that the combinatorial therapy does not appear to present significantly higher toxicities compared with sunitinib monotherapy [28]. Patients with metastatic ccRCC reported better health-related quality of life given the combination treatment compared to sunitinib [29,30].

Genetic [31,32,33,34,35,36,37,38], molecular [21,22,25,38,39,40], and clinicopathological characteristics [38,41,42,43,44] of ccRCC have not been able to fully predict clinical outcomes and prognosis of patients. RCC has distinct immunological characteristics in regard to pathogenesis and treatment, distinguishing it from other types of cancer that respond to ICI therapy. RCC harbors a relatively low mutational burden, which is expected to produce low neoantigens for antigen presentation, a situation that is often associated with a poor response to ICI therapy. Counterintuitively, RCC is known to be highly immunogenic, resulting in the infiltration of immune cells, including CD8^+^ T cells [45,46] with high cytotoxic activity [45,47]. Unlike most solid tumors, where the infiltration of CD8^+^ T cells is normally associated with a good prognosis [44], increased CD8^+^ T cell infiltration is not associated with prognosis in some studies [35,43,48,49] and actually predicted a poor prognosis in other studies [41,42,43]. Moreover, certain types of mutations that are associated with increased tumor antigen presentation and CD8^+^ T cell infiltration in most solid tumors, such as missense mutations, are not correlated with T cell infiltration in RCC [45,47,50]. The expression of immune checkpoints, such as programmed cell death protein 1 (PD-1) and programmed death-ligand 1 (PD-L1), have not been convincingly shown to predict clinical response to ICI in RCC [21,22,25,38,39,40]. Meanwhile, new characteristics have been uncovered as potential factors that enable the prediction of clinical response to ICI. For example, human endogenous lentivirus virus expression or defective antigen presentation may be a key factor for poor response to ICI in ccRCC patients [38,45]. Taken together, current basic, translational, and clinical research underscores the need to further investigate the tumor immune microenvironment in ccRCC to predict patient outcomes, to identify patients who are likely to respond to immunotherapy, and/or to determine new immunotherapy modalities to treat patients who are not responsive to current ICI therapy.

Single-cell RNA sequencing (scRNAseq) technology dissects the dynamic and heterogeneous tumor microenvironment by characterizing the transcriptome and genome at the single-cell level, providing a prominent method for painting a detailed picture of the immune landscape when studying cancer immunology [51,52]. Integrating various components of scRNAseq transcriptome into multi-omics measurements provides a better understanding of cell identity, fate, and function in the context of both normal biology and pathology [52,53]. The application of scRNAseq to renal parenchyma or kidney cancer is just at its inception and is helping provide a clearer understanding of cell of origin, tumor and immune cell heterogeneity, immune-suppressive microenvironment, therapeutic response, and ultimately prediction of prognosis [54,55,56,57,58,59,60,61,62].

Here, we summarize the landmark clinical trials for immunotherapy applied to ccRCC and translational scRNAseq research focusing on ccRCC, which is the most immunogenic subtype among RCC subtypes [56]. This review provides translational evidence and potential targets that can be utilized to improve cancer immunotherapy.

## 2. Immunotherapeutic Updates of ccRCC

### 2.1. Cytokine-Based Immunotherapy

IL-2 is a cytokine that modulates immunity and tolerance by acting on lymphoid cells, including CD8^+^ T cells, as a growth factor and activator [63]. The activation of CD8^+^ T cells facilitates the tumor-killing effect through the recognition of neoantigens presented by the tumor cells [63]. The FDA approved the usage of high-dose IL-2 (600,000 IU/kg) in metastatic RCC in 1992 based on the pooled results of several phase II studies [64,65], representing the first FDA-approved immunotherapy for RCC. These pooled results showed a 14% overall ORR, with 5% of patients having a CR and 9% having a partial response (PR). An even higher dose of IL-2 (720,000 IU/kg) was administered to metastatic RCC patients, yielding a 20% ORR and 9% CR [66]. Similar results supporting the efficacy of a higher dose of IL-2 have been reported [64,67]; and intriguingly, the favorable response of high dose IL-2 was associated with PD-L1 expression, regardless of the patients’ clinical classification [68]. High-dose IL-2 is clinically administrated with intensive care requiring an inpatient hospital stay but with a subset of responders who have extremely durable responses. Several studies have determined the efficacy of interferon-α 2a (IFNα2a) and found anti-tumor effects on patients with advanced ccRCC with an ORR of 6% to 10% [69,70,71]. The ORRs of the two cytokines are in general low in ccRCC patients, and the major hurdle for their clinical use also lies in the significant toxicities affecting multiple major organs [72,73].

### 2.2. Tyrosine Kinase and mTOR Inhibitors

Following IL-2 therapy, clinical treatment of ccRCC moved more towards the use of tyrosine kinase inhibitors (TKIs) targeting VEGFA/VEGFR pathway and neoangiogenesis, including sunitinib, sorafenib, and cabozantinib for treating ccRCC patients [74,75,76]. In 2006, sunitinib was introduced to treat metastatic RCC patients as the first-line therapy after the phase III trial showed that patients with sunitinib treatment had a significantly longer PFS, compared to those who were treated with IFNα [76]. Sorafenib was another classical TKI approved as second-line therapy for patients who had disease progression following conventional therapy for ccRCC. Treatment with sorafenib significantly prolonged the PFS in advanced ccRCC patients when compared to placebo [75]. In subsequent years, more TKI inhibitors with higher potency and more specificity, including pazopanib, cabozantinib, axitinib, and lenvatinib, were added to the treatment options for RCC patients [74,77,78,79,80].

Temsirolimus and everolimus are two inhibitors for the mammalian target of rapamycin (mTOR) that have been approved for treating RCC patients. mTOR is a highly conserved protein kinase that regulates HIFs-related metabolism and proliferation of ccRCC cells via the PI3K and Akt pathways [81,82,83]. In 2007, FDA approved treatment with temsirolimus following a phase III clinical trial in patients with metastatic RCC [84]. Patients receiving temsirolimus alone experienced longer overall survival (OS) and PFS than those who received IFNα alone. Everolimus was approved by FDA in 2009 for patients who failed sunitinib and sorafenib treatment [85], after showing clinical efficacy in patients who failed to respond to these therapies. Although numerous clinical trials and studies as described above have demonstrated the superior efficacy of TKIs to previous cytokine-based therapy, most ccRCC patients will develop acquired resistance within one year [86].

### 2.3. Immune Checkpoint Inhibitors

Currently, immune checkpoint blocking agents, including antibodies that inhibit PD-1, PD-L1, and cytotoxic T-lymphocyte-associated protein 4 (CTLA-4), are being successfully investigated and applied to the patients with ccRCC.

The first clinical trial of ICIs in ccRCC was conducted in 2007, attesting to the effect of CTLA-4 blockade in patients with metastatic RCC [87]. The phase II study included patients receiving either 3 mg/kg followed by 1 mg/kg or only 3 mg/kg of ipilimumab (anti-CTLA-4) for 3 weeks. One of the 21 patients with a lower dose and five of 40 patients with a higher dose had partial responses. There is a significant correlation between patients with autoimmune events and tumor regression, suggesting that the reinvigoration of CD8^+^ T cells promotes the tumor-killing effect. However, due to limited efficacy, the use of ipilimumab as monotherapy for RCC was halted.

A second clinical trial of ICIs in patients with ccRCC attested to the effect of PD-1 blockade on patients with ccRCC, with an ORR of 27% (9 out 33 patients) [14]. In this later phase II study, patients with metastatic ccRCC previously treated with anti-VEGF therapy were administrated 0.3, 2, or 10 mg/kg nivolumab (anti-PD-1). The median PFS was 2.7 months, 4.0 months, and 4.2 months respectively. The OS was 18.2 months, 25.5 months, and 24.7 months respectively [88]. In CheckMate 025, a phase III study, patients previously treated with anti-angiogenic therapy received either 3 mg of nivolumab or 10 mg of everolimus [15,89]. Although progression-free survival showed no difference between the two treatments, the OS for nivolumab was 25.0 months compared to 19.6 months for everolimus (*p* = 0.002) [89]. Also, the nivolumab-treated group showed a greater response rate (25% compared to 5% in the everolimus-treated group). Extended follow-up confirmed the superior efficacy of nivolumab over everolimus.

The first combination therapy was initiated in 2012, attesting to the efficacy of nivolumab with sunitinib, pazopanib, or ipilimumab [90,91]. Patients treated with nivolumab plus sunitinib showed a 55% ORR and median PFS of 12.7 months. For the group treated with nivolumab plus pazopanib, the ORR was 45% and PFS was 7.2 months. The nivolumab plus ipilimumab treatment was divided into two dose regimens: patients received either 3 mg/kg of nivolumab and 1 mg/kg of ipilimumab or 1 mg/kg of nivolumab and 3 mg/kg of ipilimumab. Both treatment regimens had an ORR of about 40% and a 2-year OS of 68%. The nivolumab group showed a lower rate of adverse events (38.3%) compared to the ipilimumab group (61.7%). In the phase III CheckMate 214 trial, the combination of nivolumab with ipilimumab was tested against sunitinib alone [21,27,92]. According to the criterion from the International Metastatic RCC Database Consortium (IMDC), intermediate and poor-risk patients receiving nivolumab + ipilimumab had a survival rate of 75% at 18-months compared to a 60% survival rate at 18 months for sunitinib. The ORR was 42% for the group treated with nivolumab plus ipilimumab, compared to 27% for the group treated with sunitinib. The CR was 9% and 1% in the combination and monotherapy, respectively. In the follow-up study, the nivolumab plus ipilimumab combination had a superior OS to the sunitinib therapy within the intermediate and poor-risk and intent to treat patients.

Because anti-VEGF treatment was found to have immunomodulatory effects on different types of immune cells, including myeloid cells and regulatory T cells (Treg) [93,94,95,96], clinical trials with the combination of ICIs and anti-VEGF agents were investigated in RCC. In an open-label phase III trial (Keynote 426), 861 patients with previously untreated advanced ccRCC were assigned to either axitinib plus pembrolizumab (anti-PD-1) or sunitinib alone group [25,97,98]. The 1-year survival rate was 89.9% for the combination group compared to 78.3% for the sunitinib alone. The median PFS for the combination treatment was also significantly higher than the sunitinib alone group (15.1 months vs. 11.1 months). The ORR was 59.3% and 35.7%, respectively. The study revealed that patients treated with axitinib plus pembrolizumab demonstrated a better response in all three IMDC risk groups, regardless of PD-L1 expression. Another clinical trial (Clear/Keynote 581) confirmed the superior efficacy of the combination of pembrolizumab (anti-PD-1) plus lenvatinib—a TKI targeting RET, KIT, PDGFR, and VEGFRs—over everolimus [26]. In this phase III trial, 1069 untreated patients with ccRCC were assigned to pembrolizumab plus lenvatinib, lenvatinib plus everolimus, or sunitinib at a 1:1:1 ratio. The ORR was 71%, 53.5%, and 36.1%, and the median PFS was 23.9 months, 14.7 months, and 9.2 months for the experimental arms of pembrolizumab plus lenvatinib, lenvatinib plus everolimus, and sunitinib, respectively. Encouraging results were also obtained in the CheckMate 9ER trial where 651 untreated patients with advanced ccRCC were assigned to treatment with either Nivolumab (240 mg every 2 weeks) plus cabozantinib (40 mg once daily)—a TKI targeting AXL, RET, MET, TIE-2, and VEGFRs—or sunitinib (50 mg once daily for 4 weeks of each 6-week cycle) [22,99]. This phase III study showed that the combination significantly improved PFS and OS as compared to sunitinib alone. At 18.1 months of median follow-up, patients who received the combination had a median of 16.6 months of PFS with a 55.7% ORR, whereas those who received sunitinib alone had a median PFS of 8.3 months and a 27.1% ORR. At 12 months, the probability of OS was higher in the combination arm (85.7%) compared to those in the control arm (75.6%). The clinical benefit of the nivolumab and cabozantinib over sunitinib was observed regardless of PD-L1 expression.

The JAVELIN Renal 101 trial compared the combination of avelumab (anti-PD-L1) plus axitinib with sunitinib alone [23,100,101]. Patients with PD-L1 positive tumors (as defined by ≥1% of immune cells immunohistochemistry (IHC)-staining positive within the tested tumor area) showed a median PFS of 13.8 months for the combination therapy compared to 7.2 months for the sunitinib alone. The ORR was 55.2% and 25.5%, respectively. This study showed that avelumab plus axitinib could be an effective therapy for patients with PD-L1 positive ccRCC. However, the follow-up study on biomarker analysis revealed that the expression of PD-L1 was not correlated with a better response and PFS in patients receiving avelumab plus axitinib [33]. Another approved combination therapy for metastatic RCC is atezolizumab (anti-PD-L1) with bevacizumab—a monoclonal antibody targeting VEGFA. In the phase III study (IMmotion 151), patients were randomly assigned to atezolizumab with bevacizumab or sunitinib alone [24,30]. The median PFS survival was 11.2 and 7.7 months for the PD-L1 positive population (as defined by ≥1% of immune cells IHC-staining positive within the tested tumor area), tested with atezolizumab plus bevacizumab or sunitinib alone, respectively. There was a difference in OS, but the patients experienced fewer treatment-related adverse events.

Altogether, based on clinical trials and publications, the clinical benefit of immune checkpoint inhibitors and their combination with anti-angiogenic agents is evident in both untreated and treated patients with advanced ccRCC. Clinically relevant results from the phase III clinical trials are summarized in Table 1.

### 2.4. Ongoing Clinical Trials

Table 2 summarizes the ongoing phase III clinical trials that cover a wide range of critical issues, including the efficacy of newly developed ICIs, the role of immune checkpoint in the previously established experimental arms, the efficacy of ICI as adjuvant therapy on the rate of recurrence following nephrectomy [104,105,106], and the effect of salvage ICI following progression on ICI treatment [107]. In addition, other studies are also testing the role of small molecules inhibitors in combination immunotherapy [108], the effect of IL-2 in combination with ICI [109], the efficacy of ICI on brain metastasis [110], and the optimal sequence of ICIs [111].

Briefly, the COSMIC-313 study is now being conducted to evaluate the efficacy of cabozantinib in combination with nivolumab and ipilimumab as the first therapy using a triplet. The study is designed to determine whether the addition of cabozantinib leads to clinical benefit over the combination of the ICIs as far as patient’s PFS and OS. PDIGREE is another clinical trial investigating the therapeutic role of cabozantinib in patients who have completed receiving nivolumab and ipilimumab therapy. PIVOT-09 is being conducted to examine the effect of bempegaldesleukin (IL-2 agonist) in combination with nivolumab versus either sunitinib or cabozantinib, and this clinical trial will compare the ORR and OS in an intermediate or poor-risk group of untreated ccRCC patients.

Another study (NCT04736706) will determine the efficacy, safety, and the specific role of belzutifan (HIF-2 inhibitor) [115] and quavonlimab (anti-CTLA-4) in combination with pembrolizumab and lenvatinib. Clinical trials of RAMPART, CheckMate 914, IMmotion010, and NCT03055013, will determine the post-surgical clinical benefit of ICIs (anti-PD1/PD-L1 and/or anti-CTLA-4) versus active monitoring in patients with partial or total nephrectomy. NCT04510597 will study the role of cytoreductive nephrectomy in combination with systemic ICI in ccRCC patients. CheckMate-67T is being conducted to study the efficacy, safety, and tolerability of nivolumab when patients are given the ICI subcutaneously. NCT04157985 will determine the optimal treatment duration of anti-PD-1 and PD-L1 therapies.

In summary, clinical evidence is sufficient to demonstrate that ccRCC is highly immunogenic and has great potential for durable response to immunotherapy. The next step is to solve the riddle of why only some patients have clinical benefits during ICI treatment, while others show intrinsic or acquired resistance to ICIs and ensuing disease progression and poor prognosis. Various molecular features of ccRCC obtained from bulk multi-omics approaches cannot precisely predict patients’ prognosis and clinical response to ICI, at least in part due to the substantial heterogeneity in immune cell contents in ccRCC. scRNAseq is the most comprehensive tool to study immune cells at the genome-wide and single-cell levels in order to uncover immune cell heterogeneity. Using scRNAseq to define the complex ccRCC immune microenvironment offers unique opportunity to elucidate potential mechanisms and/or markers for response to ICI therapy, as well as possible targets for improving response rates to ICIs.

## 3. Single-Cell Genomics to Study the Tumor Microenvironment

Single-cell genomics determines the genetic, epigenetic, or chromatin structure information at the single cell level with optimized next-generation sequencing (NGS) technologies. scRNAseq has become a potent tool to provide a higher resolution of the transcriptome for individual cells. scRNAseq can be used to study the cellular heterogeneity for given tissues to identify a rare and novel cell population that would not be detected by conventional methods, to determine cell state transitions affected by intrinsic and extrinsic stimuli, to understand differential genes/pathway alterations between cell populations, and to explore the clonal status of T or B cells when combined with T or B cell receptor sequencing, etc. [116,117]

Here we summarize published studies adopting scRNAseq technology with a focus on cancer immunology of ccRCC (Table 3). We will introduce some basic concepts and common processes of scRNAseq technology, including scRNAseq library preparation and common computational analyses. In detail, single-cell analysis technologies, including scRNAseq, and their applications in cancer immunology have been previously reviewed in detail [51,117]. Different scRNAseq library preparation methods have been reviewed [118,119]. Current studies applying scRNAseq technology to RCC have largely adopted a droplet-based platform provided by 10× Genomics. As such, we mainly focus on a droplet-based microfluidic system for scRNAseq library preparation.

### 3.1. Basic Concept and Experiment-Related Workflow of Microfluidic-Based scRNAseq

Microfluidic droplet-based scRNAseq has been used as one of the useful platforms to study single-cells in cancer immunology [118,119,120]. The droplet-based microfluidic system does not necessarily need cell sorting but needs high viability cells for preserving molecular states and reads either 3′ or 5′ end of the transcripts with barcoding and unique molecular identifier (UMI) tagging [118,119,120]. Droplet-based scRNAseq is characterized by high cellular resolution, low amplification noise, and high cost-effectiveness for the transcriptome quantification of large numbers of cells [118,119,120]. Also, it is more suitable for the identification of diverse cell types and measurement of gene expression changes between conditions [118,119,120].

The microfluidic system automates parallel sample partitioning and captures the single cells into individual oil droplets containing uniquely barcoded beads called Gel Beads-In Emulsions (GEM) [118,120]. Poly(A) tail at the 3′ end of RNA extracted from a single-cell in an individual GEM is bound to millions of the barcoded oligonucleotides with high capture efficiency and reverse transcribed to the first strand of DNA. Subsequently, a second strand synthesizing process and a PCR amplifying process are conducted to generate analysis-ready transcriptomes on a cell-by-cell basis from the complementary DNA (cDNA) libraries [120]. Illumina sequencer is widely used for library sequencing, including published ccRCC scRNAseq studies. The directed 5′ or 3′ chemistry allows for 98 base pair sequencing, limiting the mutational analysis of sequences. Cell Ranger from 10× Genomics, one of the frequently used computational pipelines for handling raw data files, provides wrapper functions that support the packages required for the raw data pre-processing pipeline [118].

After data pre-processing, including quality control, sequence alignment, and quantification of the raw sequence, a gene expression matrix is generated from the reads mapped to exon regions with high mapping quality. R toolkit Seurat has been used for the data processing, generating the Seurat object as an input file for subsequent processes [121]. Bioconductor-based workflow and Scanpy are also popular toolkits for R and python users, respectively [122,123]. Data analysis and visualization follow a standard preprocessing workflow that includes selection and filtration of cells based on quality control, data normalization and scaling, and the detection of highly variable features. The highly variable features are used for principal component analysis (PCA). After the data pre-processing steps, a high-dimensional molecular profile for individual cells is computationally classified into distinct cell populations [117,121]. Individual cells are clustered based on distances of components and visualized by non-linear dimensionality reduction techniques, such as t-distributed stochastic neighbor embedding (t-SNE) [124] or uniform manifold approximation and projection (UMAP) [125]. Although analysis varies depending on the study design, one can conduct main analyses with complementary computational techniques, such as cell composition, cell state transitions, differential gene expression, pathway analysis, cell-fate trajectories, molecular interactions, and cellular interactions [118].

### 3.2. ScRNAseq in ccRCC

The tumor microenvironment of ccRCC is extremely heterogeneous in its molecular and immune phenotypes [11,58,126,127,128]. As discussed above, means of predicting response to ICI therapy in other solid tumors have not proven clinically useful in RCC. Single-cell proteomics, as implemented by flow cytometry, mass cytometry, or multiplexed immunohistochemistry, has identified cell composition and potential cell types that generate and maintain the immune suppressive microenvironment of RCC [43,46,61]. Although these single-cell analysis technologies are useful and informative, they are inherently limited by the available number of pre-selected antibodies, resulting in the identification of only anticipated cell types [117]. The deconvolution method using bulk RNA-seq can be used to estimate immune cell composition, but this method is nowhere close to fully reflecting the heterogeneous immune composition of any tissues [127,128]. Evaluation of proliferation of CD8^+^ T cells by Ki-67 positivity has been indicated as a favorable prognostic factor [42], however, this has been contradicted by recent studies with scRNAseq results [58,59,60].

Currently, scRNAseq, which is not limited by the determining markers, has dissected tumor heterogeneity in multiple types of human solid cancers [51]. In ccRCC, a few studies with scRNAseq have just begun to investigate immune cell heterogeneity, immune pathogenesis, and response to immunotherapy [54,55,56,57,58,59,60,61,62]. Analyzing tumor-infiltrating immune cells by scRNAseq, especially focusing on T cell exhaustion, suppressive TAMs, and inhibitory cell to cell interactions has shown to have clinical prognostic and predictive value regarding clinical outcomes and the response to immunotherapy. Thus, it needs to provide evidence of the substantial potential of scRNAseq to give insights into some of the current issues regarding RCC immunotherapy. In this review, we highlight scRNAseq studies that report key events associated with the immune environment, ccRCC progression, and response to immunotherapy. Scheme and detailed information concerning scRNAseq studies applied to ccRCC is summarized in Figure 1 and Table 3.

To define the tumor-specific change in the infiltration of immune cells, our group [58] generated droplet-based scRNAseq and single-cell T cell receptor sequencing (scTCRseq) libraries and studied 25,688 cells from matched blood and primary. Tumor samples originating from 3 untreated patients diagnosed with different grades of ccRCC. We also integrated the scRNAseq data with a previous scRNAseq dataset containing 11,367 cells derived from normal renal parenchyma and blood. The study examined immune events and cell state transitions associated with a tumor-specific environment. There was a significant increase in the population of CD8^+^ T cells and macrophages in ccRCC but a decrease in the population of CD4^+^ T cells and B cells, compared to blood and non-tumor tissues. While infiltrating tumor tissue, CD8^+^ T cells showed a transcriptional continuum from naïve to activation, but eventual exhaustion with highly expanded clonotypes. A small subset of tumor-infiltrating CD8^+^ T cells were characterized by preferential cytokine signaling and associated with a favorable response to anti-PD1 therapy. In general, tumor-infiltrating CD4^+^ T cells showed a transcriptional continuum toward more activated states, such as high cytolytic and interferon activities, as previously described [60]. Meanwhile, distinct subsets of TAMs characterized by the gene expression associated with either chemo/cytokines, apolipoproteins, or DC-like, showed high plasticity between pro- and anti-inflammatory phenotypes. Using machine-learning training with the Cancer Genome Atlas (TCGA) RCC cohort, we developed unique gene signatures defining either a subset of proliferative CD8^+^ T cells or a subset of DC-like TAMs. Both scRNAseq signatures had a prognostic value of predicting a poorer prognosis in the OS of patients with ccRCC. Using external mass cytometry data [46], we also confirmed the existence of the proliferative CD8^+^ T subset as a PD1^+^Ki-67^hi^ phenotype in ccRCC. Supporting the scRNAseq-based prognostic model, the PD1^+^Ki-67^hi^ CD8^+^ T cells are highly enriched with co-stimulatory proteins and immune checkpoints, such as ICOS, 4-1BB, TIM-3, CTLA-4, HLA-DR, and CD38.

Zhang et al. [56] identified the peculiar immune environment and pathogenesis of ccRCC. The study analyzed 29,131 cells derived from adjacent non-tumor tissues and primary ccRCCs from 9 patients. In addition to identifying the putative cell of origin for ccRCC, the study evaluated the potential source of immune infiltration to ccRCC and the prognostic value of distinct cell populations. Supporting the previous scRNAseq study applied to ccRCC [57], a subset of proximal tubular cells and neoplastic epithelial cells were predicted to recruit immune cells to tumor site via IFN response, including especially secretion of serine protease C1s. This is further supported by a positive correlation between the degree of TAM fraction and the C1S gene expression in bulk RNA-seq, scRNAseq, and TCGA RCC datasets. Two different subsets of TAMs, defined by chemokine/cytokine- versus lysosome-related genes, had dichotomous prognostic values of predicting OS within the same TCGA RCC cohort. Using bulk RNA-seq obtained from metastatic ccRCC patients who were treated with TKI followed by anti-PD1 therapy, the study defined genes associated with clinical benefit. Notably, endothelial cells and pericytes predominantly expressed the genes negatively associated with the response, and genes associated with clinical benefit were primarily expressed among T cells. In TCGA ccRCC dataset, however, treatment-naïve patients with a high fraction of endothelial cells in localized ccRCC were predicted to have better OS. Patients with a high estimated fraction (>90th percentile) of either tumor-infiltrating CD8^+^ T cells or plasmalemma vesicle associated protein (PLVAP)^+^ endothelial cells were separately present in the scatter plot, suggesting mutual exclusivity of the two cell types concerning clinical outcome in the ccRCC environment.

Obradovic et al. [61] also identified and characterized the tumor-specific immune environment of ccRCC using scRNAseq data. The study studied 163,905 cells isolated from adjacent non-tumor tissues and primary, non-metastatic ccRCC from six untreated patients. Moving beyond mRNA expression, the study applied a specific algorithm, called VIPER, to scRNAseq data and inferred single-cell protein activity. Of note, the VIPER-based protein activity inference turned out to significantly overcome challenges of scRNAseq, including recovery of transcriptome dynamics masked by dropouts up to 70% to 80%, and thus was able to precisely predict single-cell protein activity. This was also validated by using flow cytometry and an external CITE-seq dataset. The integrated analysis enabled the identification of potentially targetable novel master regulatory proteins in a rare population that would have been undetectable by gene expression-based analysis. VIPER analysis led to the identification of ccRCC-infiltrating exhausted CD8^+^ T cells, Treg, TAMs, and CD45^-^ cell types with a high resolution. The protein activity of the C1Q family of proteins, APOE, and TREM-2 was significantly upregulated in macrophages in tumors compared to non-tumor tissues. VIPER was also successful in obtaining the inferred protein activity from bulk RNA-seq data derived from untreated ccRCC surgical resections. In two independent cohorts, the VIPER-applied protein signature of tumor-specific macrophages was not only preferentially enriched in patients who underwent post-surgical ccRCC recurrence but also significantly associated with the shorter time-to-recurrence in the Kaplan–Meier curve. The representative leading-edge proteins among TAM-defining markers were APOE and TREM-2. Using multiplexed immunohistochemistry, C1Q^+^TREM-2^+^ TAMs were found to be tumor-specific and C1Q^+^TREM-2^+^APOE^+^ TAMs located significantly nearer the tumor cells than triple-negative TAMs. The proximity was also strengthened by the analysis of ligand-receptor interaction between tumor cells and APO^+^ TAMs. The frequency of either C1Q^+^ or TREM-2^+^ TAMs was higher in tumor slide sections from patients with recurrence than those with non-recurrence. Clinically, the density of C1Q^+^ TAMs above a certain threshold of 0.01 was significantly associated with ccRCC recurrence.

To define the change in the infiltration of immune cells with advancing ccRCC, Braun et al. [60] generated droplet-based scRNAseq and scTCRseq libraries and analyzed 164,722 cells isolated from blood, adjacent non-tumor tissues, and different stages of primary and metastatic ccRCC from 13 untreated patients. As RCC progressed from early to locally advanced and metastatic diseases, there was a consistent increase in the frequency of terminally exhausted CD8^+^ T cells, Treg, CD14^+^ monocytes, and immune suppressive M2-like TAMs, and a general decrease in the frequency of cytotoxic CD8^+^ T cells, central memory CD4^+^ T cells, and inflammatory M1-like TAMs. Pseudotime analysis coupled with gene signature also confirmed the progressive dysfunction and exhaustion of tumor-infiltrating CD8^+^ T cells with advancing ccRCC. Likewise, the trajectory analysis showed preferential enrichment of pro-inflammatory and anti-inflammatory scRNAseq signatures in earlier-stage and metastatic-stage ccRCC, respectively. Ligand-receptor interactions were inferred to tumor-infiltrating immune cells. Intriguingly, while a majority of non-exhausted T cells in earlier-stage ccRCC were predicted to have few interactions, terminally exhausted CD8^+^ T cells in advanced ccRCC were inferred to have numerous ligand-receptor pairs within the myeloid populations, including TAMs. With metastatic ccRCC samples, the inhibitory interaction between two populations was further supported by the multiplexed immunofluorescence-based spatial proximity and upregulated expression of ligands and their cognate receptors. Using multiple external ccRCC datasets [46,129,130], the authors also showed a significant increase in the proportion of terminally exhausted CD8^+^ T cells and M2-like TAMs and the gene signature score defining the inhibitory interaction with the advancing ccRCC stage. The high expression of the gene signature was specifically associated with poor prognosis in the OS of patients with late-stage ccRCC. Meanwhile, the gene signature did not have prognostic value for predicting PFS and immune response to anti-PD1 therapy or mTOR inhibitor. On the other hand, scTCRseq results showed a significant decrease in the TCR diversity with advancing ccRCC stage, and there was a high proportion of terminally exhausted CD8^+^ T cells with low TCR diversity in metastatic ccRCC. Contrary to the previous finding [62], the shared clonotypes were preferentially detected in tumors rather than non-tumor tissues.

To identify potential immune populations that drive the response to ICI, Krishna et al. [55] collected 167,283 cells from blood, adjacent non-tumor tissue, metastatic lymph node, and multiple regions of primary ccRCC from 2 untreated and 4 treated patients with ICI. Then, the authors generated droplet-based scRNAseq and scTCRseq libraries. First of all, multiregional sampling confirmed extensive heterogeneity within and between patients, highlighting the vulnerability of applying bulk RNA-seq-derived signatures to tumor region sampling bias. Mapping the immune environment of ccRCC identified diverse immune cell types, such as five well-defined CD8^+^ T clusters and 4 clusters of TAMs characterized by HLA or ISG expression. Next, the authors co-analyzed scRNAseq and pathologic review, and identified that tissue-resident CD8^+^ T cells, as well as CD4^+^ T cells and NK cells, were heavily infiltrated in tumor regions associated with tumor regression or CR to ICI. Conversely, in tumor regions associated with resistance to ICI, a high proportion of HLA^+^ TAMs were identified with a scarcity of T cells. Following ICI treatment, the tissue-resident CD8^+^ T cells from the complete responders were found to solely undergo clonal expansion with unique TCR clonotypes, but the resistant non-responders also had the clonal expansion of the CD8^+^ T subset. To estimate potential immune populations underlying ccRCC patient prognosis and response to ICI and TKIs, various clinical signatures, such as T effector, angiogenesis, and myeloid inflammation, were applied to multiple external ccRCC cohorts [37,78,128]. Results indicated that effector T cells and angiogenic myeloid cells had the potential to elicit a favorable response to anti-PD-L1 and TKI arms. Also, the scRNAseq signature of ISG^high^ TAMs was highly associated with angiogenesis in the TKI arm. In the end, the study validated the scRNAseq signatures that are highly specific for tissue-resident CD8^+^ T cells or ISG^high^ TAMs, and applied them to IMmotion 150/151 (anti-PD-L1 plus anti-VEGF or TKI), JAVELIN Renal 101 (anti-PD-L1 plus TKI) ccRCC cohorts. Importantly, high levels of the tissue-resident CD8^+^ T signature were significantly associated with improved PFS and better response in anti-PD-L1 and TKI arms. Autologously, the ISG^high^ TAMs signature was significantly associated with improved PFS in the TKI arm. However, both signatures did not predict clinical outcomes from the TCGA ccRCC dataset.

In a similar study of four anti-PD-1-treated patients and three untreated patients with primary and metastatic ccRCC, Bi et al. [59] generated a droplet-based scRNAseq library and analyzed 34,326 cells. The study started off applying progenitor or terminally exhausted signature to the scRNAseq immune subsets and identified 4-1BB^low^CD8^+^ T cells that resembled the progenitor exhausted population, which is known to persist long term, respond to anti-PD1 therapy, and ultimately differentiate into the terminally exhausted population in melanoma [131]. Following ICI treatment, the 4-1BB^low^CD8^+^ T cells were found to upregulate the expression of effector and co-stimulatory molecules, including GRANZYME A (GZMA) and FAS LIGAND (FASLG), and highly enriched with terminally exhausted signature. This result was also supported by the high enrichment score of 4-1BB-low signature in PD1-exposed CD8^+^ T cells from the CheckMate 009 cohort. Similarly, ICI treatment rendered all distinct subsets of TAMs more M1-like and pro-inflammatory in responder patients, at least in part, as induced by IFN secreted from CD8^+^ T cells. At the same time, however, the ICI-exposed 4-1BB^low^CD8^+^ T cells and TAMs also showed systemic and dramatic upregulation of immune checkpoint and evasion genes, suggesting progressive and eventual acquisition of ICI resistance. Two subsets of cancer cells identified were found to transcriptionally shift toward a pro-inflammatory state during ICI. Patients who had a high score of the gene signature that defined renal morphogenic and angiogenic cancer population showed the ICI-specific clinical benefit regarding OS in the CheckMate 025 cohort (anti-PD1 arm). Supporting different cell populations in a complex cross-talk in ccRCC environment, numerous ligand-receptor pairs, including IFNγ-producing CD8^+^ T cells and type 2 IFN receptor on TAMs, were inferred and further supported by expression signatures and estimated immune cell fractions adapted from CheckMate 009 cohort.

Very recently, Au et al. [38] scrutinized key determinants that are responsible for clinical response in metastatic ccRCC patients before and after nivolumab treatment. Again, various tumor molecular features of ccRCC, including single mutations, copy number alterations, insertion-and-deletions, mutational burden, and neoantigen load, were not associated with favorable anti-PD-1 response. Of note, however, ccRCC-specific expression of human endogenous retrovirus was found to be associated with lack of response to nivolumab. In addition, it has been suggested that defects in antigen presentation, despite a high number of mutations resulting from defective DNA mismatch repair, might be a potential factor underlying poor response to ICI. Authors generated droplet-based scRNAseq and scTCRseq libraries and analyzed a total of 25,456 IgG4+ (anti-PD-1 antibody-bound) and IgG4- CD3 T cells isolated from a responder and a non-responder during nivolumab monotherapy. scRNA-seq showed that anti-PD-1 treatment renders nivolumab-bound ccRCC-infiltrating CD8^+^ T cells immunologically activated in both responder and non-responder. Paired analysis of scRNAseq and scTCRseq found that nivolumab treatment induces clonal expansion of pre-existing CD8^+^ T cells, and only the responder had clonal hyper-expansion of the nivolumab-bound CD8^+^ T cells (as defined by more than 200 clones with the same complementary determining region 3 sequence). The expanded nivolumab-bound CD8^+^ T cells had higher expression of GZMK gene in the responder than the non-responder. scRNAseq, flow cytometry, and multiplexed IHC confirmed the higher expression of GZMB and TCF7 in the nivolumab-bound CD8+ T cells from responders. Using previously published ccRCC-specific scRNA/scTCRseq datasets, they further validated their findings. As a result, expanded TCRs in responders but not the non-responders had higher expression of genes involved in T cell activation and co-stimulatory markers, including GZMK and 4-1BB. It should also be noted that nivolumab treatment not only reinvigorated CD8^+^ T cells in the responder, but also caused T cell exhaustion and dysfunction, suggesting simultaneous development of resistance as consistent with the previous finding [59]. Finally, bulk and scTCRseq analysis before and after treatment demonstrated that responders have clonal expansion of pre-existing and novel TCRs from the nivolumab-bound CD8^+^ T cells. However, non-responders had an overall paucity of expanded pre-existing TCRs, rather showing clonal replacement of expanded TCRs. The novel expanded T cell clones after nivolumab treatment were not associated with clinical response.

Meanwhile, several studies have also been reported using droplet-based scRNAseq technology to provide insight into normal and ccRCC immunobiology. Yu et al. [54] studied the inter-tumoral heterogeneity using bilateral ccRCC samples within a patient and identified the high similarity of the gene expression between the immune cells in the bilateral ccRCC. Liao et al. [132] mapped the atlas of single-cells that normally reside in healthy renal tissues, providing the reference data for normal renal cell biology and kidney disease. Besides the major analysis that identifies cancer cell identity, Young et al. [57] highlighted the VEGF signaling circuit in the ccRCC environment. The study identified that TAMs, as well as ccRCC cells, were a further source of VEGF, and VEGFR was highly expressed in ascending vasa recta endothelial cells. Using multiple types of human cancers, including ccRCC, Wu et al. [62] showed that expanded clonotypes from effector-like CD8^+^ T cells were simultaneously detected in the tumor, non-tumor tissues, and peripheral blood. In particular, further evidence indicated that peripherally expanded T cells with ICI treatments were directly linked to tumor infiltration and eliciting an immune response, rather than reinvigorating the already exhausted T cells in the tumor environment. This study also identified distinct subsets of immune cells with a focus on T cells in ccRCC, but did not fully characterize tumor microenvironment. Kim et al. [133] compared and analyzed scRNAseq data generated from tumor cells isolated from the patient’s metastatic ccRCC and the paired primary and metastatic ccRCC derived from the patient-derived xenograft (PDX) model. The study verified the current patient’s drug refractoriness, identified candidate signaling pathways and drugs, and validated the predicted drug sensitivity using in vitro and in vivo assays, suggesting the clinical applicability of scRNAseq and combined mouse model to screen optimal choice of TKIs.

### 3.3. Major Immune Cell Types Associated with Poor Prognosis and Resistance to ICIs

The paradox where high infiltration of CD8^+^ T cells is not linked to favorable prognosis and response to ICI in patients with ccRCC stems from the existence of exhausted and/or dysfunctional T cells. Indeed, the exhaustive status is shown to limit the actual effector function of the ccRCC-infiltrating CD8^+^ T cells [58,59,60]. The exhaustive phenotype of the T cells is being overlapped by several groups, as characterized by upregulation of PD-1, LAG-3, TIM-3, CTLA-4, TOX, and CD39 [36,46,55,58,59,61]. scRNAseq studies identified the association between the exhausted and/or dysfunctional CD8^+^ T cells and disease progression and/or resistance to ICI in patients with ccRCC. Supporting this, the exhausted T cells are unlikely to be fully reversed and reinvigorated by ICI during ccRCC treatments as suggested in other cancers [134,135,136,137,138].

The skewed polarization of TAMs toward M2-like or anti-inflammatory properties is a common feature of advanced ccRCC. Some TAM phenotypes have been reported to decrease the overall immune temperature of the ccRCC. For example, TAMs characterized by high expression of HLA are shown to promote resistance to ICI [55]. A subset of TAMs characterized by a high level of immune regulatory genes, such as APOE, C1Q, and TREM-2, has been commonly identified in the human ccRCC and RENCA model [55,58,60,61,139]. This subset is shown to be associated with a poor prognosis of ccRCC patients due to disease recurrence [61]. Complement activation and/or metabolic reprogramming can be key events associated with TAMs that shape the immunosuppressive tumor microenvironment of ccRCC [55,58,60,61,139]. 

Computational analysis using a repository of curated receptors, ligands, and their interactions enabled the identification of interactions between malignant and non-malignant cells in ccRCC [140,141]. There are multiple interactions reported between terminally exhausted CD8^+^ T cells, M2-like/anti-inflammatory TAMs, and ccRCC cells via numerous pairs of ligands and their cognate receptors (Figure 2) [55,56,59,60,61]. The inhibitory circuit becomes significant as the disease progresses, which promotes an immune-suppressive tumor microenvironment. The signature related to these interactions is found to predict a worse overall prognosis but not a response to ICI of ccRCC patients [59]. Following ICI treatment, immune checkpoint and evasion genes, such as LGALS9 and NECTIN2 expressed on tumor cells as well as TAMs, may play a role in the acquired ICI resistance [59].

Treg cells are one of the important immune-suppressive cell types. Tumor-infiltrating Treg cells are highly immunosuppressive to effector cells. Most scRNAseq datasets have a relatively low abundance of Treg cells for ccRCC, one of the reasons that Treg cells are much less focused from the aforementioned scRNAseq datasets. scRNAseq analysis has identified the increase in the frequency of the tumor-infiltrating Treg cells with advancing ccRCC [60]. Patients showing CR to ICI have low Treg infiltration by scRNAseq [55]. We have particularly focused on tumor-infiltrating Treg cells from our own ccRCC dataset [58]. Comparing tumor-infiltrating versus blood Treg cells, we identified some common shared signature genes of tumor-infiltrating Treg cells, including some genes whose protein products are targetable such as CD177 and BCL2L1 (encoding BCL-X_L_). Tumor-infiltrating Treg cells exhibit certain heterogeneity including two distinct populations, with one population showing strong suppressive capacity. We developed a unique tumor-infiltrating Treg cell signature with the prognostic value superior to some known Treg signatures [142]. The clinical importance of tumor-infiltrating Treg cells has been correlated with poor prognosis and response to immune perturbation in other studies as well [45,95,143]. A study observed that anti-PD-1 therapy induces hyper-progression with clonal expansion of tumor-infiltrating Treg cells with upregulation of some genes, including CD177 and BCL2L1 in a leukemic patient [144], the two genes we found to be elevated specifically in tumor-infiltrating Treg cells. CD177 is a surface protein and may modulate the immune suppressive function and maintain homeostasis of tumor-infiltrating Treg cells in ccRCC [142]. We have demonstrated that CD177^+^ tumor-infiltrating Treg cells are hyper-suppressive to effector T cells and anti-CD177 antibody is able to block the suppressive function of CD177^+^ tumor-infiltrating Treg cells. Our group has been actively developing other ways of targeting human tumor-infiltrating Treg cells to induce the degradation of BCL-X_L_ using proteolysis-targeting chimera (PROTAC), which seems very effective for inducing anticancer immunity [145]. Taken together, Treg cells are a potential cell type that can be targeted for cancer immunotherapy.

### 3.4. Limitations and Challenges in scRNAseq Technology

Accumulating scRNAseq studies have provided a tremendous amount of critical information that can help to solve the current issues, such as low efficacy and resistance to ICI in patients with ccRCC. Nevertheless, there are limitations and challenges in this scRNAseq technology. In general, the sample sizes are small due to the cost associated with scRNAseq. It is of the utmost importance to prepare freshly isolated single cells for the successful generation of the cDNA library [51]. Single-cell suspension with less than 70% of cell viability is not recommended for library preparation. A highly collaborative work setting is needed for prompt sample preparation and processing to secure cell viability. There is a high economic burden and upfront cost because drop-based scRNAseq platforms require expensive hardware and preparatory kits. Cell hashing and multiplexing technology where oligo-tagged antibodies against ubiquitously expressed surface proteins uniquely label cells from biologically different samples are expected to decrease costs [146]. Processing of the raw data to generate analyzable data form, scRNAseq data requires computing systems with high memory capacity. For example, the 10× Genomics Cell Ranger requires 64 gigabytes of RAM, up to 1.5 terabytes of disk space, and a Linux-based system. Newer alignment tools, such as Alven [147] or kallisto-bustools [148], cut these system requirements by an order of magnitude. The bioinformatic analysis of scRNAseq data is still challenging; in-depth analysis of the data requires experience in coding, which can be a barrier of entry for laboratories. There is still no standard guideline for processing workflow from quality control to determining resolution and dimensionality [149].

In addition to limitations concerning the bottlenecks in implementation, there are also challenges associated with scRNAseq technology. scRNAseq is invariably limited by the dropout phenomenon where up to 93% of the count matrix can be zeros [149,150]. From the immune perspective, this dropout effect, coupled with the use of a highly-variable gene approach, makes annotating cell types and discovering small immune populations difficult [151]. A certain type of immune cells can be more susceptible to dropout. Indeed, there is a preferential dropout of transcription factors and cytokines, making CD4^+^ T cell annotation difficult [61,152]. Application of a specific algorithm to scRNAseq data to infer protein activity [61] or impute RNA values [153], at least in part, may overcome the dropout. In addition, changes in the generation of cDNA, e.g., through the adoption of the second-strand synthesis option, may also be advantageous in the recovery of cytokine and transcription factor expression [152]. Single-cell sequencing requires the generation of single-cell suspensions, leading to induction of specific genetic programs and loss of spatial information [154]. Platforms for spatial scRNAseq are emerging and will offer insights into cell-to-cell communications [155]. Unlike flow cytometry with established markers for antigen experience or cellular ontogeny, the scRNAseq toolkits are not as well-stocked. In terms of the latter, scRNAseq-based lineage tracing, using cellular tagging or mitochondrial variations, may offer a chance to look at the compartment-specific immune response [156,157]. The chemistry used to generate the cDNA libraries in scRNAseq utilize short 5′ or 3′ reads, limiting the assessment of mutational status, single-nucleotide polymorphisms, or alternative splicing, such as CD45RA versus CD45RO isoforms, which all play a role in the immune response. Recent improvements in scRNAseq chemistry may reduce this issue by generating longer cDNA sequences [158].

## 4. Perspectives and Clinical Implications

### 4.1. Consensus in Nomenclature

There is no doubt that utilizing scRNAseq technology to clinical samples enables the better dissection of tumor microenvironment of ccRCC or other cancers, providing insight into various types of immune cells that are critical for either shaping immune-suppressive environment or driving a favorable immune response following ICI. The big picture of immune cell composition can be painted at a much higher resolution than what traditional bulk RNAseq or flow cytometry have been provided, along with the gene expression data of individual immune cells. As we discussed about different studies related to the nomenclature of distinct cell subsets, it becomes evident that the field is far away from achieving consensus based on signature gene expression. As ccRCC enters the immunotherapy era, elevation in tumor-infiltrating CD8^+^ T cells, though they have been known as a bad prognosis before immunotherapy became the standard frontline treatment, provides an immune-hot microenvironment for ICI to work. Although most studies borrowed signatures based on melanoma studies to determine the nature of CD8 clusters, different studies used different nomenclatures. A similar situation applies to other major immune cell types including CD4^+^ T cells and macrophages. Based on publications and after carefully comparing different populations, CD8^+^ T cells from ccRCC have the three major populations as in melanomas, including the naïve like, cytotoxic, and dysfunctional [159], as well as a relative consensus on the proliferative and tissue-resident memory (TRM) populations. Apparently, the dysfunctional group consists of a series of populations at different and likely continuous functional stages that could be the potential targets of ICIs, with a 4-1BB^low^ cluster showing feature of progenitor exhausted phenotype and can be expanded by ICIs for cancer cell killing [55,59]. This 4-1BB^low^ CD8^+^ T could be a similar population identified in another study as TRM as both populations exhibit the expression of intermediate immune checkpoints, effector/activation molecules and likely CD44 and CD103 [55,59] that are used to define TRM cells. A clear understanding of these populations should be based on the integration of these datasets and will be able to direct the prediction of patients who may benefit from ICIs.

TAMs are another major focus on ccRCC studies with 2–5 sub-clusters from different studies. The nomenclature for TAMs can be misleading since quite a few studies still used M1-like and M2-like names to define the subtle difference of their M1 or M2 signatures. Nearly all studies did not show a distinct separation of M1- versus M2-like TAMs that rather secrete M1 and/or M2 cytokines at various levels. Several studies used the marker genes such as HLA, interferon signaling genes (ISG), other lead genes or cluster numbers to define and imply functional differences. It is clear that TAMs are very important in the pathogenesis of ccRCC and can be the major predictor for the sensitivity to ICIs. The clearer designation of different TAM clusters is important for using these TAM-related signatures for clinical predictions. 

### 4.2. ScRNAseq Reveals Mechanisms of Immune Activation

The major action of ICIs in melanoma is to rejuvenate pre-existing exhausted CD8^+^ T cells, a well-accepted mechanism of action for ICI-based cancer immunotherapy. Recent development in the field identified a potential novel mechanism by ICI-induced clonal replacement, i.e. the replacement of old CD8^+^ T cell clones with new clones from blood or adjacent normal tissues. Clonal expansion of ccRCC-infiltrating non-exhausted CD8^+^ T cells and/or de novo introduction of peripherally expanded CD8^+^ T cells to tumor site can be a more convincing and potential mechanism underlying the immune response to ICI than the widely presumed reinvigoration of the pre-existing exhausted CD8^+^ T cells [44,55,62,135,137,138,160,161]. In agreement with this notion, a recent study [38] clearly demonstrated that the diversity of pre-existing CD8^+^ T cell clones, likely those similar to 4-1BB^low^ or TRM populations identified from other studies [55,59], are critical for eliciting the favorable response within nivolumab-treated ccRCC patients. Nivolumab maintains and expands these pre-existing CD8 T cell clones to elicit an effective anti-tumor immune response. In non-responders, clonal expansion of exhausted CD8^+^ T cells [55] and expanded CD8^+^ T cells with novel TCRs are not associated with clinical response to nivolumab in ccRCC patients [38]. This novel mechanism of action makes it critical to identify the diversity of pre-existing CD8^+^ T cell clones within tumor microenvironment and to set up a threshold using deep learning to predict patient responses to ICIs. Figure 3 illustrates the current concept of immunotherapy driving clinical response to ICI in patients with ccRCC.

The presence of distinct subsets of immune suppressive and/or pro-angiogenic TAMs is believed to lead to ccRCC progression and inhibit the immune response to ICI. Potential mechanisms of action include inhibitory cell-to-cell communications, modulation of complement activation and/or metabolic reprogramming [55,56,58,59,60,61,139]. Machine-learning based algorithm has the capacity to identify the potential cell-cell interactions and TAMs process many interactions with cancer cells and other immune cells (Figure 2) to facilitate cancer progression in late stage of ccRCC patients by either directly promoting angiogenesis and/or cancer cell aggressiveness, or by indirectly inducing a more immune-suppressive network. Currently there is no effective treatment to eliminate or inhibit these TAMs, but scRNAseq-based research has defined certain populations that can be shaped by ICIs in responders where ICIs induce a more M1-like responses at the same time upregulating several immune checkpoints such as VSIR, VSIG4, PD-L2, and SIGLEC10 [59]. The function of these immune checkpoints is yet-to-be validated whether they can induce ICI resistance, but if confirmed, following treatment regimens should involve in antibodies targeting those novel checkpoints.

Another complexity comes from the interactions between essential components within ccRCC involving cancer cells, immune cells and others. An oversimplified version is shown in Figure 2 where many ligand/receptor pairs exist and can potentially induce complex cellular interactions. How can we use the identified and known information to extract the dominant signaling pair that can be interrupted? For example, as many as 14 pairs of interaction are identified between CD8^+^ T cells and TAMs including PD-1/PD-L1 pair that may dominate the immune-suppressive responses within responders treated with anti-PD-1/PD-L1 antibodies. The question is whether we can develop testing and bioinformatics pipeline for clinical treatment selections rather than treating all patients with the same drugs that are known to have relatively low responses rate.

### 4.3. Conclusions Remarks

Current scRNAseq studies have been limited by the small patient cohorts and the lack of experimental validations at functional levels. Can therapeutic intervention cause the hypothesized immune modulation in TME within patients’ tumors? Future work will be required to longitudinally address the characteristics of highly effective T cells against ccRCC in different perspectives, such as stem cell-like, metabolic, transcriptional, and epigenetic states [44]. The standardization of experimental methods, such as scRNAseq studies pooling clinical trials and in vitro or in vivo preclinical perturbation models will be required to address the effect of blocking immune checkpoints or key inhibitory molecules on the reinvigoration of exhausted T cell function, replacement of exhausted T cells by non-exhausted effector T cells, or shifting anti-inflammatory TAMs to pro-inflammatory ones [133,138,139]. Multi-omics approaches to the ccRCC environment, including spatial transcriptomics and proteomics, may reveal new gene signatures and molecular targets that reflect a functional immune niche or escape [44]. Further studies are warranted to evaluate other, less-characterized cell types, such as antigen-presenting cells or regulatory T cells, to identify novel therapeutic targets that address immune dysfunction in ccRCC [33,40,43,55,59,60,139,162,163,164].

## Figures and Tables

**Figure 1 cancers-13-05856-f001:**
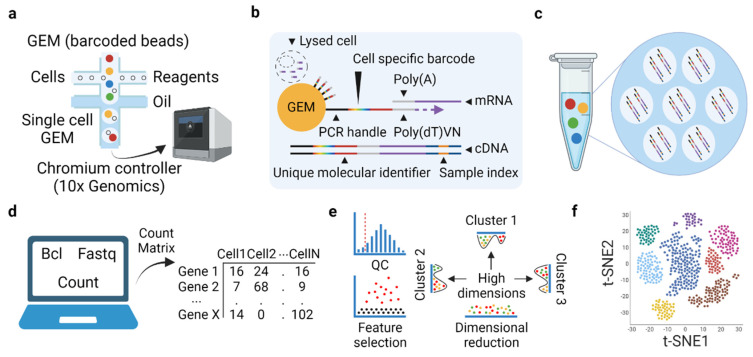
Scheme of droplet-based scRNAseq and standard bioinformatics pipeline. (**a**) Single cells are loaded to a microfluidic system and encapsulated to an oil droplet to generate single-cell GEM. (**b**) RNA released from the lysed single-cell is captured by barcoded oligonucleotides and reverse transcribed to the first and second strands of DNA. (**c**) PCR amplifying process is conducted to generate cDNA library, which is sequenced by Illumina sequencer. (**d**) Cell Ranger from 10× Genomics provides raw data pre-processing pipeline, resulting in the generation of a gene expression matrix. (**e**) Standard pre-processing steps for scRNAseq data. Low-quality cells are removed. Highly variable features are selected and used for principal component analysis. A high-dimensional molecular profile for individual cells is computationally classified into distinct cell populations. (**f**) Individual cells are visualized by non-linear dimensionality reduction techniques, such as t-SNE. Abbreviation: GEM; Gel Beads-In Emulsions, t-SNE; t-distributed stochastic neighbor embedding, cDNA; complementary DNA.

**Figure 2 cancers-13-05856-f002:**
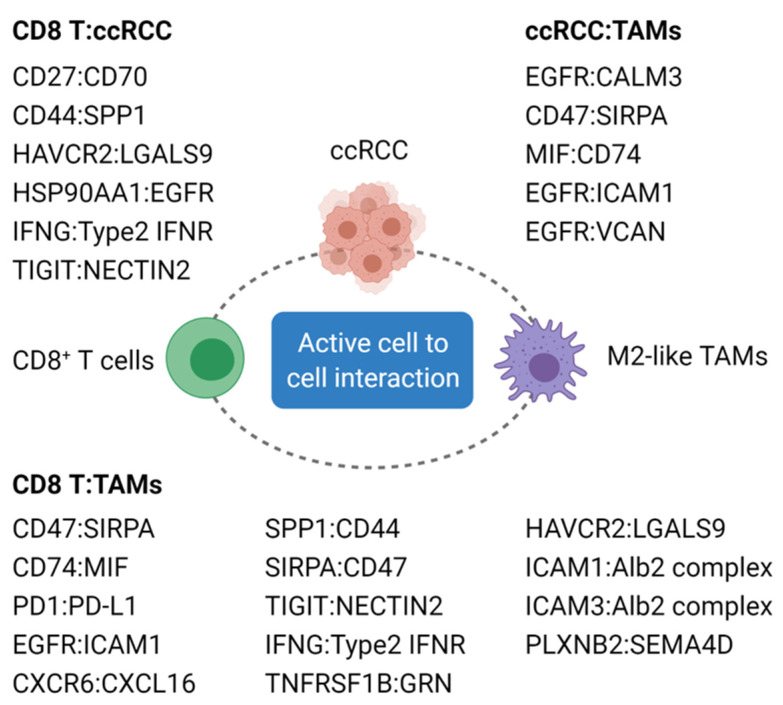
Ligand and receptor pairs potentially associated with ccRCC progression and resistance to ICI. A potential list of ligand and receptor pairs that are predicted by in silico analysis and commonly identified by published ccRCC scRNAseq studies are present. The cell to cell interaction via ligand and receptor pair has not been validated at the functional level. Abbreviation: ccRCC; clear cell renal cell carcinoma, TAMs; tumor-associated macrophages.

**Figure 3 cancers-13-05856-f003:**
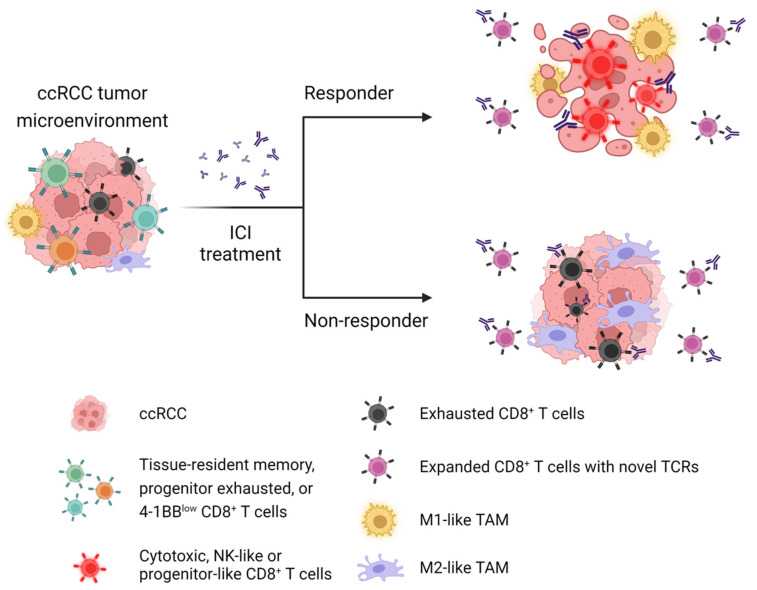
Current concept of immunotherapy driving clinical response to ICI in patients with ccRCC. Pre-existing CD8^+^ T cell clones phenotyped by CD69+ZNF683+ TRM, progenitor exhausted, or 4-1BB^low^ are considered to have a critical role in favorable response to ICI in ccRCC patients. In responders, ICI-bound expanded CD8^+^ T cells exhibit cytotoxic, NK-like, or progenitor-like phenotypes. In contrast, non-responders had no clonal expansion of the tumor-reactive CD8^+^ T cell clones. In both responders and non-responders, pre-existing exhausted T cells are clonally expanded following ICI treatment. In ccRCC, clonal expansion of CD8^+^ T cells with novel TCRs are not associated with clinical response to ICI. Following ICI treatment, TAMs shift toward M1-like or pro-inflammatory phenotype in responders, whereas non-responders have skewed polarization of TAMs toward M2-like or anti-inflammatory phenotype in ccRCC tumor microenvironment. CD69, ZNF683, and CD103 are commonly expressed in CD8^+^ TRM cells. 4-1BB^low^ CD8^+^ T cells are highly enriched with progenitor exhausted signature. The tumor-reactive effector-like CD8^+^ T cells commonly express GZMA, GZMB, GZMK, PRF1, IFNG, NKG7, CCL3, CCL5, and CXCL13 genes, as well as co-inhibitory receptors, such as PD-1, TIM-3, LAG3, and TIGIT genes. Terminally exhausted phenotype is characterized by high expression of PD-1, LAG-3, TIM-3, CTLA-4, TOX, and CD39. M1-like TAMs are highly enriched with signatures of interferon signaling, antigen presentation, and proteasome function. M2-like TAMs are commonly characterized by high expression of HLA, APOE, C1QA, and TREM-2. Abbreviation: ccRCC; clear cell renal cell carcinoma, ICI; immune checkpoint inhibitor, TAMs; tumor-associated macrophages, NK; natural killer, TCR; T cell receptor.

**Table 1 cancers-13-05856-t001:** Updated phase III clinical trials investigating immunotherapies for advanced ccRCC.

Study Name	Identifier	Agent	Target	Total	ORR	TRAE 3+	Citations
CheckMate 025 *	NCT01668784	Nivolumab	PD-1	821	23%	19%	[15,89]
CheckMate 214	NCT02231749	Nivolumab	PD-1	1096	39.1%	47.9%	[21,27,102]
		Ipilimumab	CTLA-4				
IMmotion 151	NCT02420821	Atezolizumab	PD-L1	915	37%	40%	[24,30]
		Bevacizumab	VEGF				
JAVELIN Renal 101	NCT02684006	Avelumab	PD-L1	886	52.5%	71.2%	[23,101,103]
		Axitinib	RTK				
CLEAR	NCT02811861	Pembrolizumab	PD-1	1069	71%	82.4%	[26]
		Lenvatinib	RTK				
Keynote 426	NCT02853331	Pembrolizumab	PD-1	861	60.4%	66.4%	[25,97,98]
		Axitinib	RTK				
CheckMate 9ER	NCT03141177	Nivolumab	PD-1	651	56.6%	75.3%	[22]
		Cabozantinib	RTK				

* This study used Everolimus as a control arm. Other studies used Sunitinib as a control arm. Abbreviation: ORR; objective response rate, TRAE; treatment-related adverse event, RTK; receptor tyrosine kinase.

**Table 2 cancers-13-05856-t002:** Ongoing phase III clinical trials investigating immunotherapies for advanced ccRCC.

Study Name	Identifier	Agent	Target	Control
COSMIC-313	NCT03937219 [108]	Nivolumab	PD-1	Nivolumab and Ipilimumab
		Ipilimumab	CTLA-4	
		Cabozantinib	RTK	
na	NCT03729245 [109]	Bempegaldesleukin	IL-2 agonist	Sunitinib
		Nivolumab	PD-1	Cabozantinib
Keynote 564	NCT03142334 [105]	Pembrolizumab	PD-1	Placebo
Contact 03	NCT04338269 [107]	Atezolizumab	PD-L1	Cabozantinib
		Cabozantinib	RTK	
IMmotion 010	NCT03024996 [106]	Atezolizumab	PD-L1	Placebo following nephrectomy
PDIGREE	NCT03793166 [111]	Nivolumab	PD-1	Nivolumab following Nivolumab and Ipilimumab
		Cabozantinib	RTK	
CheckMate 914	NCT03138512 [104]	Nivolumab	PD-1	Placebo following nephrectomy
		Ipilimumab	CTLA-4	
PROSPER	NCT03055013 [112]	Nivolumab	PD-1	Monitoring after nephrectomy
CheckMate 920	NCT02982954 [113]	Nivolumab	PD-1	This clinical trial examines the safety of ICI in RCC patients with either brain metastasis or Karnofsky Performance Status 50–60%
		Ipilimumab	CTLA-4	
na	NCT04736706	Pembrolizumab	PD-1	Pembrolizumab and lenvatinib
		Quavonlimab	CTLA-4	
		Lenvatinib	RTK	
		Belzutifan	HIF2	
na	NCT04523272	TQB2450	PD-L1	Sunitinib
		Anlotinib	RTK	
na	NCT04394975	Toripalimab	PD-1	Sunitinib
		Axitinib	RTK	
na	NCT03873402	Nivolumab	PD-1	Nivolumab
		Ipilimumab	CTLA-4	
RAMPART	NCT03288532 [114]	Durvalumab	PD-1	Monitoring after nephrectomy
		Tremelimumab	CTLA-4	
CheckMate 67T	NCT04810078	Nivolumab	PD-1	This clinical trial examines the safety and efficacy of subcutaneous Nivolumab injection
PROBE	NCT04510597	Nivolumab	PD-1	This clinical trial examines the efficacy of cytoreductive nephrectomy in combination with ICI
		Pembrolizumab	PD-1	
		Axitinib	RTK	
		Avelumab	PD-L1	
na	NCT04157985	Nivolumab	PD-1	This clinical trial examines the length of treatment with ICI.
		Pembrolizumab	PD-1	
		Ipilimumab	CTLA-4	
		Atezolizumab	PD-L1	

Abbreviation: ICI; immune checkpoint inhibitor, RTK; receptor tyrosine kinase, na; not applicable.

**Table 3 cancers-13-05856-t003:** scRNAseq studies identifying and characterizing immune environment associated with ccRCC progression and response to ICI.

Patient Number	Control Group	Experimental Group	Cell Number	Platform	Citation
3	PB	ccRCC	25,688	10× Genomics droplet-based	[58]
11	ANT	ccRCC	163,905	10× Genomics droplet-based	[61]
9	ANT	ccRCC	29,131	10× Genomics droplet-based	[56]
13	ANT	Advance stages of ccRCC	164,722	10× Genomics droplet-based	[60]
8	Primary and metastatic ccRCC (LN), ICI-untreated	Primary and metastatic ccRCC (LN, lung, abdomen), ICI-treated	34,326	10× Genomics droplet-based	[59]
6	ANT and primary ccRCC, ICI-untreated	PB, ANT, and multi-regions of primary and metastatic ccRCC (LN), ICI-treated	167,283	10× Genomics droplet-based	[55]
2	PB and multi-regions of primary ccRCC, ICI-untreated	PB and multi-regions of primary and metastatic ccRCC (adrenal gland, bone, nephrectomy bed), ICI-treated	26,456	10× Genomics droplet-based	[38]

Abbreviation: scRNAseq; single-cell RNA sequencing, ICI; immune checkpoint inhibitor, LN; lymph node, ccRCC; clear cell renal cell carcinoma, PB; peripheral blood, ANT; adjacent non-tumor tissue.

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
