# Peer review of "Updates on Immunotherapy and Immune Landscape in Renal Clear Cell Carcinoma"

_cancers, 2021, doi:10.3390/cancers13225856_

Round 1

Reviewer 1 Report

The review is comprehensive, well-researched, with great images and tables. It represents a timely and needed contribution to the evolving field of therapies for renal cell carcinoma.

Prior studies showed that the genetic, molecular and clinicopathologic characteristics of clear cell renal cell carcinoma (ccRCC) are not able to entirely predict clinical outcomes and prognosis. The current article demonstrates the unique immunologic features of ccRCC, including known high immunogenicity, despite harboring a relatively low mutational burden, increased infiltration by CD8+ lymphocytes, and expression of immune checkpoints that are not predictive of clinical response to therapy. All these findings raise the possibility the ccRCC tumor microenvironment may play a significant role in modulating the course of the disease and response to therapies, therefore allowing the identification of patients that respond or develop resistance to treatment.

The article is composed of two parts. The first is a thorough review of the clinical trials for immunotherapy to date, showcasing the clinical benefit of immune checkpoint inhibitors in combination with anti-angiogenic agents in management of ccRCC. Significant findings of each trial are presented in detail. The tables report the results of completed clinical trials, as well as ongoing ones, providing a great, most up to date resource for a wide variety of clinicians and researchers. The second part presents the technique of single-cell genomics, and the major contributions to identification of unique features of ccRCC tumor microenvironment. Specifically, the presence of a subset of CD8+ lymphocytes with exhaustive phenotype, and the polarization of tumor associated macrophages (TAMs) toward an anti-inflammatory immunoprofile are described, among others. For each underlying mechanism, receptor-ligand pairs with potential impact of progression and resistance to immune checkpoint inhibitor therapies are listed.

The publication of this review article is recommended and no significant revisions are needed. The identification through scRNA seq of multiple new potential mechanisms and targets raises the question of what next steps are to be taken for the new advancements’ generalization and implementation. A more detailed presentation of the possible protocols to be employed in research settings and the approach to integration in clinical practice would be beneficial. This may include statements regarding the overall availability of the technique, with implications to validation of results performed on relatively small patient cohorts.  

Author Response

1st reviewer’s comments

Authors’ Response to the comments: Thank you for the comments. We greatly appreciate the recognition of the significance.

Reviewer 2 Report

This study addresses a current topic.

The manuscript is quite well written and organized. 

Figures and tables are comprehensive and clear.

The introduction explains in a clear and coherent manner the background of this study.

We suggest the following modifications:

  • Introduction section: although the authors correctly included important papers in this setting, we believe a couple of studies should be cited within the introduction (PMID: 32799582; PMID: 27855702), only for a matter of consistency. We think it might be useful to introduce the topic of this interesting study.
  • The authors should expand some sections, including a more personal perspective to reflect on. For example, they could answer the following questions – in order to facilitate the understanding of this complex topic to readers: what potential does this study hold? What are the knowledge gaps and how do researchers tackle them? How do you see this area unfolding in the next 5 years? We think it would be extremely interesting for the readers.

One additional little flaw: the authors could better explain the limitations of their work, in the last part of the Discussion.

We suggest the addition of some references for a matter of consistency. Moreover, the authors should better clarify some points.

Author Response

Comment #1: Introduction section: although the authors correctly included important papers in this setting, we believe a couple of studies should be cited within the introduction (PMID: 32799582; PMID: 27855702), only for a matter of consistency. We think it might be useful to introduce the topic of this interesting study.

Comment #2: The authors should expand some sections, including a more personal perspective to reflect on. For example, they could answer the following questions – in order to facilitate the understanding of this complex topic to readers: what potential does this study hold? What are the knowledge gaps and how do researchers tackle them? How do you see this area unfolding in the next 5 years? We think it would be extremely interesting for the readers.

Authors’ Response to the comments: We added personal perspective in the last part of conclusion section and changed the section title to “Conclusion and perspectives” (line 730-797). We added comments on perspectives related to the use of scRNAseq to define cell populations and/or signatures that can better predict ICI therapy.

Comment #3: One additional little flaw: the authors could better explain the limitations of their work, in the last part of the Discussion. We suggest the addition of some references for a matter of consistency. Moreover, the authors should better clarify some points.

Authors’ Response to the comments: We appreciate your suggestion and added the suggested references. We made thorough clarification throughout the manuscript. We added limitations and challenges of scRNAseq in the last part of the discussion (line 684-729).

Reviewer 3 Report

In their manuscript "Immunotherapeutic updates and immune landscape shaped by single-cell RNA sequencing in clear cell renal cell carcinoma", Kim et al. summarize the history and current developments / trials in the treatment of clear cell renal cell carcinoma (RCC). Moreover, they summarize the current evidence regarding single-cell RNA sequencing as a novel technology in RCC. Especially the last topic appears interesting as basis for a review. Their manuscript is well written and appears scientifically sound. 

Regarding weaknesses of the draft:
Given the current structure of the manuscript, it is unclear how the two parts of this draft (1: clinically established / approved therapies in RCC; 2: novel single-cell RNA sequencing approaches) fit together. Both parts are not sufficiently linked.

In addition, the text suffers from a lack of illustrations - with only one figure describing single-cell RNA sequencing technology - instead of illustrating for example high-risk or ICI resistance constellations.

Minor points:

  • line 63: from a clinical perspective, this sentence appears misleading.
  • line 68: combination therapies in 2009.
  • line 123: abbreviation ORR (CR etc.) should be introduced earlier (in chapter 1)
  • line 146: lenvatinib
  • Table 2: check layout; additionally: study names missing - not linked to the text body.
  • line 373: concerning.

Author Response

Comment #1: Given the current structure of the manuscript, it is unclear how the two parts of this draft (1: clinically established / approved therapies in RCC; 2: novel single-cell RNA sequencing approaches) fit together. Both parts are not sufficiently linked.

Authors’ Response to the comments: We have added the rationale in the last part of therapeutic section (lines 291 to 302) and also connected those two fields by expanding significantly the last section (4. Perspectives and Clinical Implications).

Comment #2: Regarding weaknesses of the draft: In addition, the text suffers from a lack of illustrations - with only one figure describing single-cell RNA sequencing technology - instead of illustrating for example high-risk or ICI resistance constellations.

Authors’ Response to the comments: Great points and we added two more figures Figure 2 and Figure 3. We can add more illustrations if the reviewer thinks of any important content.

Comment #3: line 63: from a clinical perspective, this sentence appears misleading

Authors’ Response to the comments: We made the sentence being more concise (Page2: Lines 63 to 65).

Comment #4: line 68: combination therapies in 2009

Authors’ Response to the comments: We corrected the sentence.

Comment #5: line 123: abbreviation ORR (CR etc.) should be introduced earlier (in chapter 1)

Authors’ Response to the comments: We introduced the abbreviation of ORR (line 77) and CR when they appeared first time (line 77).

Comment #6: line 146: lenvatinib

Authors’ Response to the comments: We corrected the typo (line 153).

Comment #7: Table 2: check layout; additionally: study names missing - not linked to the text body.

Authors’ Response to the comments: We included study name if available and changed the layout to be clear. Also, new reference was included for RAMPART (NCT03288532) in Table 2.

Comment #8: line 373: concerning.

 Authors’ Response to the comments: We corrected the typo (line 392).

Round 2

Reviewer 2 Report

The authors modified the paper according to our suggestions.

We recommend Acceptance.

Reviewer 3 Report

Comments were adequately addressed - Accept.